

# Impact of litter quantity on the soil bacteria community during the decomposition of *Quercus wutaishanica* litter

Quanchao Zeng[1,2,*], Yang Liu[1,*] and Shaoshan An[1,2]

[1] College of Natural Resources and Environment, Northwest A&F University, Yangling, China
[2] State Key Laboratory of Soil Erosion and Dryland Farming on the Loess Plateau, Northwest A&F University, Yangling, China
[*] These authors contributed equally to this work.

## ABSTRACT

The forest ecosystem is the main component of terrestrial ecosystems. The global climate and the functions and processes of soil microbes in the ecosystem are all influenced by litter decomposition. The effects of litter decomposition on the abundance of soil microorganisms remain unknown. Here, we analyzed soil bacterial communities during the litter decomposition process in an incubation experiment under treatment with different litter quantities based on annual litterfall data (normal quantity, 200 g/(m$^2$/yr); double quantity, 400 g/(m$^2$/yr) and control, no litter). The results showed that litter quantity had significant effects on soil carbon fractions, nitrogen fractions, and bacterial community compositions, but significant differences were not found in the soil bacterial diversity. The normal litter quantity enhanced the relative abundance of Actinobacteria and Firmicutes and reduced the relative abundance of Bacteroidetes, Plantctomycets and Nitrospiare. The Beta-, Gamma-, and Deltaproteobacteria were significantly less abundant in the normal quantity litter addition treatment, and were subsequently more abundant in the double quantity litter addition treatment. The bacterial communities transitioned from Proteobacteria-dominant (Beta-, Gamma-, and Delta) to Actinobacteria-dominant during the decomposition of the normal quantity of litter. A cluster analysis showed that the double litter treatment and the control had similar bacterial community compositions. These results suggested that the double quantity litter limited the shift of the soil bacterial community. Our results indicate that litter decomposition alters bacterial dynamics under the accumulation of litter during the vegetation restoration process, which provides important significant guidelines for the management of forest ecosystems.

# INTRODUCTION

Plant litter is the main source of soil carbon and nitrogen, and influences the function and development of terrestrial ecosystems (*Sauvadet et al., 2016*). The interaction between the soil and plant litter microorganism has attracted much attention (*Urbanová, Šnajdr*

Corresponding authors
Yang Liu, roshanlx@163.com
Shaoshan An, shan@ms.iswc.ac.cn

*& Baldrian, 2015*). Microorganisms provide the link between the soil and plant and plays an important role in the soil biogeochemical recycle, including the recycling of carbon (C), nitrogen (N), phosphorus (P) and other mineral elements (*Keiluweit et al., 2015*). Plants are the major sources of soil nutrients and affect soil properties via litter decomposition, root exudates and microorganism invasion from litter (*Wardle et al., 2004*). Litter decomposition is a key process for element recycling and had been studied by many researchers in different areas (*Aerts, 1997*; *Fanin, Hättenschwiler & Fromin, 2014*; *Freschet et al., 2013*; *Gundel et al., 2016*; *Kuramae et al., 2013*; *Sauvadet et al., 2016*; *Van Huysen, Perakis & Harmon, 2016*). Previous studies have shown litter quality and quantity are the main factors that drive the litter decomposition process (*Keiluweit et al., 2015*). Litter quality includes the C, N, P, Mn, Fe, Ca, Al, cellulose, hemi-cellulose and lignin content in the litter (*Aerts, 1997*; *Berg & Mcclaugherty, 2014*; *Keiluweit et al., 2015*). Litter represents a major pathway for C cycling between the vegetation and the soil in terrestrial ecosystems, and changes in the aboveground litter quantity and quality could have important consequences for C cycling. Some researchers have reported that litter quantity increased litter decomposition, litter carbon (C) loss and soil respiration, but did not alter soil organic carbon content after 2.5 years in the forest system (*Fang et al., 2015*). Generally, the total C and N contents of soil is not sensitive to the litter decomposition process, but soil organisms have proved to be a sensitive indicator of the response of vegetation restoration (*An et al., 2013*; *Huang et al., 2011*). The quality of litter inputs determines on both the genetic structure of the soil microbial communities and their substrate use patterns, which may have effects on soil microbial structure (*Lamarche et al., 2007*; *Zhang et al., 2013*). Thus, much more attention should be paid to the response of sensitive soil indicators to litter decomposition with the increase of the litter layer.

With the on-going Grain for Green project in China that began in 1999, plant coverage, plant biomass and the litter layer have gradually increased on the Loess Plateau (*Deng, Liu & Shangguan, 2014*). Enhanced soil quality and soil carbon storage have been reported by many researchers (*An et al., 2013*; *Cheng et al., 2015*; *Deng, Shangguan & Sweeney, 2013*). With the process of vegetation restoration, plant litters gradually accumulate, which may influence the function of soil microorganisms. Litter quantity is a key factor that can influence the function and composition of soil organisms. Higher plant litter quantities usually favor the growth of opportunistic bacterial taxa for the greater labile C compounds from litter (*Nemergut et al., 2010*). Thus, the accumulation of plant litter should theoretically enhance the biomass of soil microbes, in particular, organisms better that are suited to address the greater availability of C compounds via exploitative resource strategies (*Nemergut et al., 2010*). However, the relative effects of litter quantity on the soil bacterial structure have rarely been assessed, and to our knowledge, there are no studies disentangling the effects of litter quantity on the soil bacteria during the decomposition processes in forest soils.

With the objective of disentangling the effects of litter quantity on soil bacterial structure and function, we analyzed the soil community structure and diversity in an incubation experiment with different litter quantities, including normal and double levels based on the data from annual litter fall. Illumina Hiseq sequencing was used to determine the

response of the soil bacterial community to different amounts of litter decomposition. We hypothesized that: (1) litter decomposition may enhance the soil bacterial diversity and community composition, especially for the oligotrophic bacteria, and (2) this trend will increase with the increase of litter quantity as more nutrients are available from litter decomposition. Our results provide insights to better understand the process of litter decomposition and to manage forest land with accumulated plant litter.

## MATERIALS AND METHODS

### Site description

Soil and litter samples were collected from the Fuxian Observatory for Soil Erosion and Eco-environment, a secondary forest region. *Quercus wutaishanica* was the predominant community, playing an important role in maintaining the stability of the system in this area (*Fan, Wang & Guo, 2006*; *Guo et al., 2010*). Therefore, understanding the effects of *Quercus wutaishanica* leaf litter decomposition provides insights into the carbon and nitrogen recycling in the soil-plant system. We established three plots in *Quercus wutaishanica* forests with similar topographical conditions to investigate the annual litter fall using the method described by *Ukonmaanaho & Starr (2001)*. Over two years of observations, the annual litter fall of Quercus wutaishanica was approximately 200 g/m$^2$/yr.

### Soil and litter sampling

Soil samples from 0–20 cm were obtained in September 2015 when most of the leaves had fallen. All roots, stones, small animals and other debris were removed from the soil samples by hand, and the samples were sieved through a 2 mm screen. The mixed soils were used to conduct the litter decomposition experiment in the laboratory. The soil organic carbon and total nitrogen contents were 18.26 g/kg and 1.60 g/kg, respectively. Fresh litter was collected with a litter collector. To avoid damaging the litter structure, the leaves were air-dried for more than two weeks at room temperature to a consistent weight.

### Litter decomposition experiment

Litter decomposition experiment was conducted using the nylon mesh bag technique. There were three treatments, including normal quantity (200 g/(m$^2$/yr)) litter, double quantity (400 g/(m$^2$/yr)) litter, and control (no litter) (Fig. 1). The litter bags (10 cm × 20 cm size) were constructed out of 1 mm nylon mesh. First, we placed 200 g fresh soils in a 1 L plastic basin and then placed a litter bag (5 g, normal quantity; 10 g, double quantity) on the surface. Each treatment had three replicates. We also conducted a control experiment without litter bags. All basins were incubated at 25 °C in an incubator. The soil water content was adjusted using the weighting method every week at a relative humidity of 20%. After 90 days, we collected the soil sample layer below the litter bags to analyze the soil properties and bacterial communities. After harvest, each soil sample was mixed and separated into two parts. One part was air-dried for the evaluation of the soil properties. The other part was frozen at −80 °C (using liquid nitrogen) for subsequent sequencing analysis.

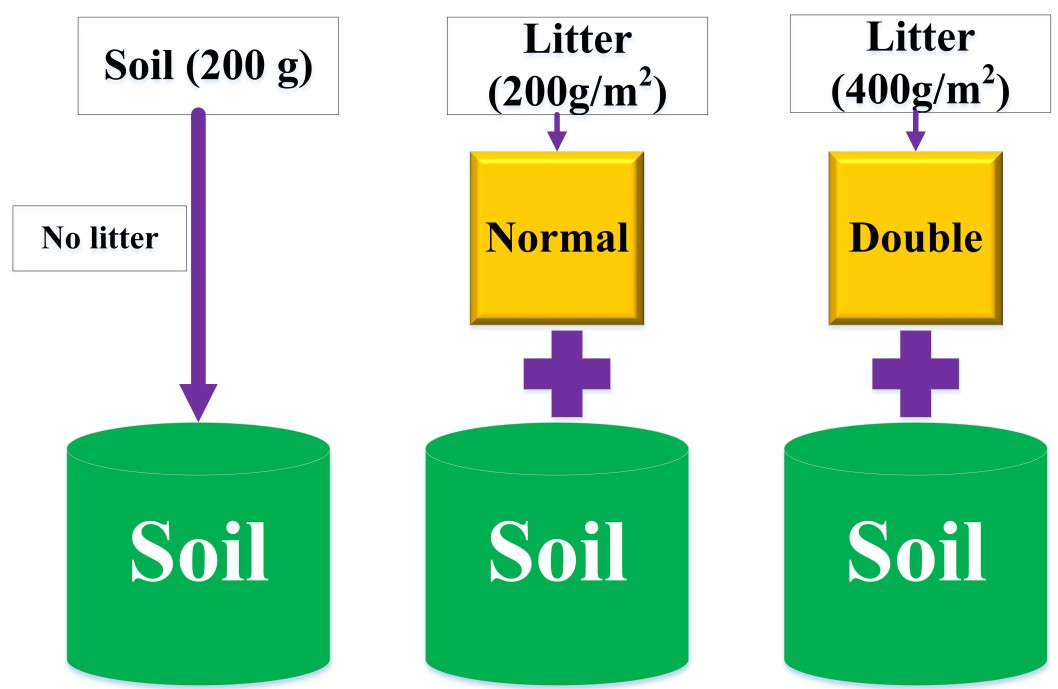

**Figure 1** The setup of the litter decomposition experiment under different litter quantities.

## Analysis of the soil properties

The soil moisture was determined gravimetrically with fresh soils at 105 °C for 24 h, and the water content was expressed as a percentage of the dry weight. The fumigation-extraction method was used to determine microbial biomass carbon (MBC) and microbial nitrogen (MBN) (*Vance, Brookes & Jenkinson, 1987*). The dissolved carbon (DOC) and dissolved nitrogen (DON) in the soil were determined by extracting the samples in 0.5 mol/L $K_2SO_4$. The soil total N (STN), soil organic carbon (SOC), soil nitrate nitrogen ($NO_3^-$-N) and soil ammonia nitrogen ($NH_4^+$-N) were analyzed using the method described by *Zeng et al. (2016)*.

## Soil NDA extraction and PCR amplification

The DNA of the soil was extracted from a 0.5 g soil sample using the CTAB method. The concentration and purity of the DNA were monitored using 1% agarose gels. According to the concentration, the DNA samples were diluted to 1 ng/μL with sterile water to reduce the effects of the PCR inhibitors. The V4 gene of the 16S rRNA was amplified using the 515F/806R primer sets (*Bergmann et al., 2011*; *Zeng, An & Liu, 2017*). All PCR reactions were carried out with Phusion® High-Fidelity PCR Master Mix (New England Biolabs, Ipswich, MA, USA). The same volume of 1 × loading buffer (contained SYB green) was mixed with the PCR products, and electrophoresis was conducted on 2% agarose gels for detection. The samples with a bright strip between 400–450 bp were chosen for further experiments. The PCR products were mixed at equal density ratios. Then, the PCR mixtures were purified using a Qiagen Gel Extraction Kit (Qiagen, Hilden, North Rhine-Westphalia, Germany).

## Illumina Miseq sequencing

Sequencing libraries were generated using a TruSeq® DNA PCR-Free Sample Preparation Kit (Illumina, Hayward, CA, USA) following the manufacturer's recommendations and index codes were added. The library quality was assessed on the Qubit@ 2.0 Fluorometer (Thermo Scientific, Waltham, MA, USA) and Agilent Bioanalyzer 2100 system. Finally, the library was sequenced on an Illumina HiSeq 2,500 platform and 250 bp paired-end reads were generated. The 16S rRNA gene amplicon sequencing was conducted at Novogene Bioinformatics Technology Co., Ltd., Beijing, China. The raw sequence data in FASTQ format are accessible from the NCBI SRA with the number of SRP107086.

## Statistical and bioinformatics analysis

QIIME software was used to analyze the sequences data (*Caporaso et al., 2010*). The sequencing data yielded 569,171 raw reads, with 71,146 raw reads per sample. After removing the low quality reads and trimming the barcodes and primers, there were 545,740 valid reads (average length 253 bp). Clustering sequences at 97% similarity levels were assigned to the same OTUs (*Stackebrandt & Goebel, 1994*). After the removal of chimeric sequences, a total of 4,833 different OTUs were recorded. Taxonomy was assigned to each OTU via the Ribosomal Database Project (RDP) classifier (*Cole et al., 2009*). The representative sequence for each OTU was screened for further annotation. The abundance of OTUs information was normalized using a standard sequence number corresponding to the sample with the fewest sequences. The alpha diversity was applied to analyze the complexity of the species diversity of each sample, including the observed-species index and the Shannon index. All indices in our samples were calculated with QIIME (Version1.7.0) and displayed with R software (Version 2.15.3).

The similarities between treatments were measured using a principal coordinate analysis (PCoA) plot. The PCoA was analyzed using the WGCNA, stat and ggplot2 packages in R software (Version 2.15.3). One-way ANOSIM and SIMPER analysis were used to compare the differences in the bacterial composition among the different treatments using the Bray-Curtis method (PRIMER software v 7) (*Zeng, An & Liu, 2017*). A higher R value in ANOSIM indicated a higher separation between the treatments. The linear discriminant analysis effect size (LEfSe) method was used to determine the difference between the normal and the double litter amount treatments (*Segata et al., 2011*). One-way ANOVA was performed to explore the differences between the soil properties and the soil bacterial compositions under the different treatments (SPSS version 20.0 for Windows), and the Student-Newman-Keuls (SKN) method was used for the comparison ($P = 0.05$). The relationships between soil bacterial composition and the environmental factors were tested using Pearson correlation analyses using SPSS 20.0 for Windows.

# RESULTS

## Soil chemical properties and the response of microbial biomass to litter decomposition

The soil nitrogen fractions, carbon fractions and soil moisture were significantly altered by the addition of litter (Fig. 2). The soil moisture showed a significant decline in the normal

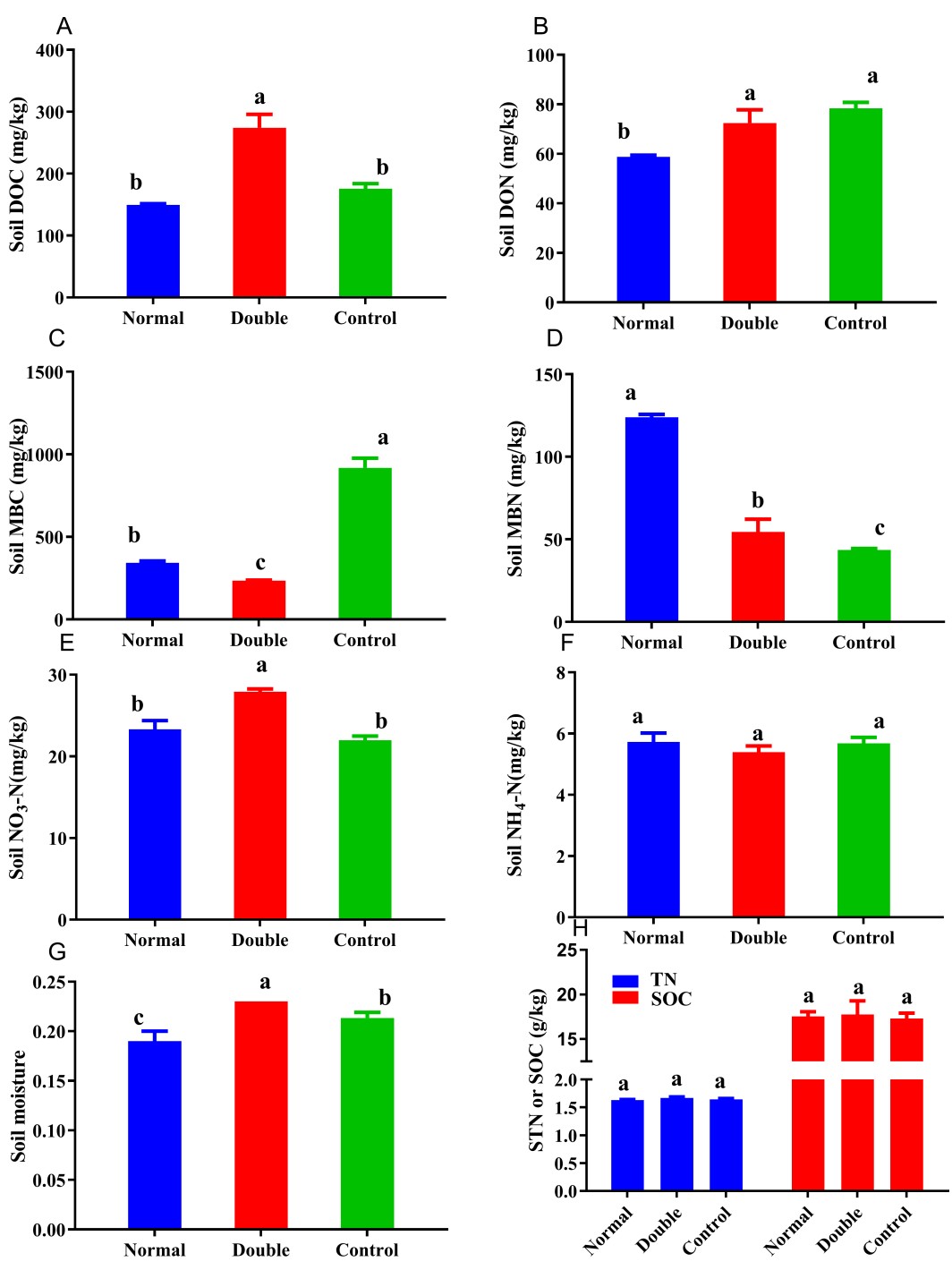

**Figure 2  Soil carbon and nitrogen fractions in the different treatments.** Different lowercase letters indicate significant differences at the 0.05 level. All data are expressed as means ± SD.

**Table 1  Soil bacterial alpha diversity indices under different the litter quantity treatment.**

| Treatment | Observed_species | Shannon |
| --- | --- | --- |
| Normal | 3,035 ± 42 | 9.57 ± 0.11 |
| Double | 2,962 ± 109 | 9.59 ± 0.04 |
| Control | 2,932 ± 62 | 9.53 ± 0.10 |

**Notes.**

All indices were not significantly different between the different treatments. All data are expressed as means ± SD.

treatment and an increase in double treatment. No significant differences were observed among the treatments for soil $NH_4^+$-N, which ranged from 5.39 to 5.73 mg/kg. The MBN content was significantly higher in the normal treatment and ranged from 43.50 to 124.14 mg/kg, and was in the order of normal>double >control. The DON showed the opposite trend to the MBN, with the highest value measured in the control treatment. The soil nitrate nitrogen ranged from 21.98 to 27.90 mg/kg, and there was no significant difference between the normal and the control treatments. The control treatment had the highest MBC and the lowest DOC, and was significantly different from the double treatment. With the increase of litter quantity, the soil nitrate nitrogen, soil moisture, MBC, DOC and DON showed significant reductions in the normal treatment, and a significant increase was observed in the MBN.

## Response of the soil bacterial community activity to litter decomposition

The bacterial diversity indices showed no significant changes between the different treatments (Table 1), but the soil bacterial community compositions demonstrated significant structuring in response to litter addition. The most dominant groups across all soil samples were Proteobacteria (38–42%), Actinobacteria (11–21%), Acidobacteria (18–20%), Gemmatimonadetes (5%), Bacteroidetes (4–6%), Chloroflexi (3%), Firmicutes (1–2%), Verrucomicrobia (2–4%), Planctomycetes (3–4%) and Nitrospirae (2%) (Fig. 3). The relative abundance of Actinobacteria, Bacteroidetes, Planctomycetes, Firmicutes and Nitrospirae in the normal treatment was significantly higher than in the double and control treatments (Fig. 3A).

To explore the dynamics of the major microbial taxa under different mounts of litter treatment, we found that Alpha, Beta, Gamma, and Delta-proteobacteria were the main members of Proteobacteria. Only Alpha-proteobacteria showed no significant differences among the different treatments, and ranged from 15.50 to 17.82%. With the increase of litter quantity, the relative abundance of Bet, Gamma, and Deltaproteobacteria showed a decrease in the normal treatment, and an increase in the double treatment. The Beta, Gamma, and Deltaproteobacteria occupied 5.75%, 6.00%, and 6.93%, respectively, in the normal treatment, which significantly differed from the double and control treatment (Fig. 3B). At the order level, Subgroup_6 and Subgroup_4 were the dominant taxa in the Acidobacteria phylum, and showed no significant changes with the increase of litter quantity. Rhizobiales was the dominant taxa of Alpha-proteobacteria, and ranged from 7.01 to 8.75%, and showed similar variation to those of the Alpha-proteobacteria. Solirubrobacterales, Xanthomonadales, Sphingobacteriales, Myxococcales and Gaiellales

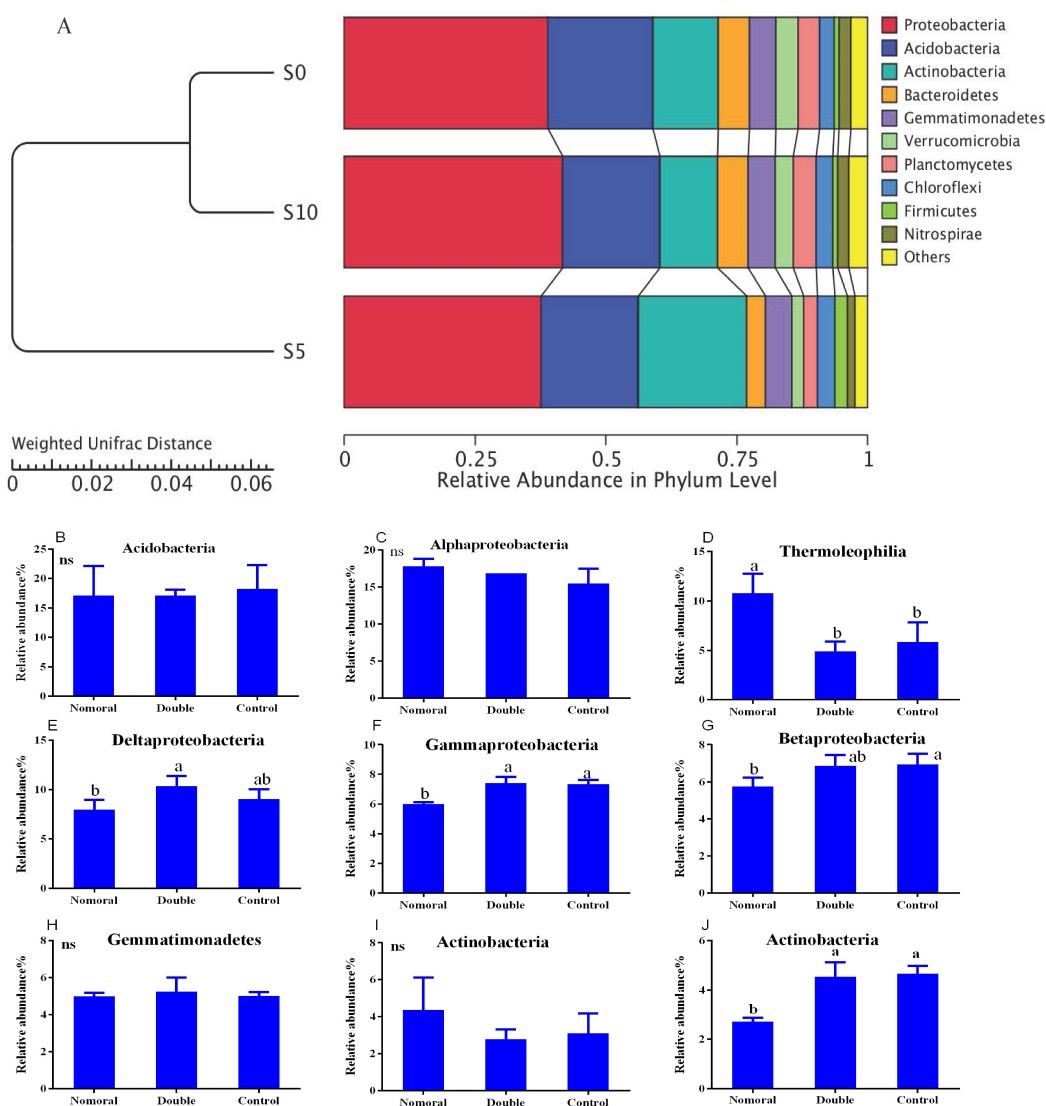

**Figure 3   Soil bacterial communities under different litter quantities at the phylum level (A) and class level (B).** Different lowercase letters indicate significant differences between the different litter quantity treatments ($P < 0.05$); ns indicates that there is no significant difference. All data are expressed as means ± SD. S0, control; S5, normal treatment; S10 double treatment.

had significant differences among the litter addition treatments (Fig. 4). These differences were only detected between the normal treatment and the double or the control treatment. The cluster analysis and PCoA also indicted these changes (Figs. 3 and 5). More specifically, the bacterial community profiles in normal treatment trended to group together and were separated from those in the double and control treatments. A $t$-test showed that the soil bacterial taxa were significantly different between the normal and the double treatments, including Proteobacteria (Xanthomonadales, Salinisphaerales, Legionellales, Chromatiales, Syntrophobacterales, Sh765B-TzT-29, Myxococcales, SC-I-84, Sneathiellales, DB1-14 and

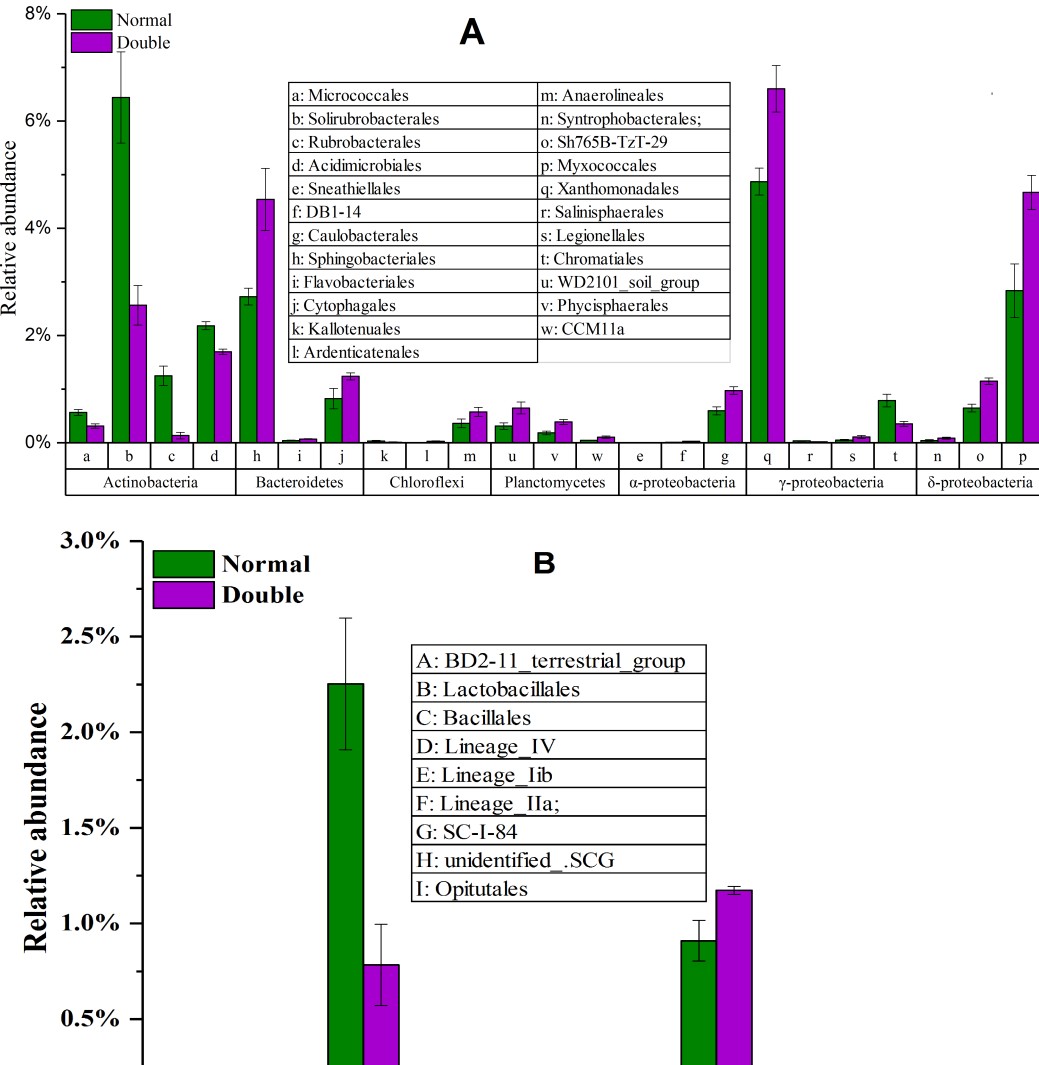

**Figure 4** **The significantly different taxa between the normal treatment and the double treatment as determined by a *T*-test.** The taxa shown in the figure were significant at the 0.05 level. All data are expressed as means ± SD.

Caulobacterales), Planctomycetes (WD2101_soil_group, Phycisphaerales, CCM11a), and Actinobacteria (Micrococcales, Solirubrobacterales, Rubrobacterales and Acidimicrobiales) (Fig. 5).

The ANOSIM based on the OTUs of the 16S rRNA gene sequences indicated that the differences were significant between the different litter addition treatments (ANOSIM Global $R = 0.761$, $P = 0.01$). SIMPER analysis revealed that bacterial communities were 76–81% similar between the normal, double and control treatments. The LEfSe analyses identified the significant difference in the abundant taxa between the different litter quantity treatments. Using the LEfSe, we found that Bacteroidetes, Myxococcales and

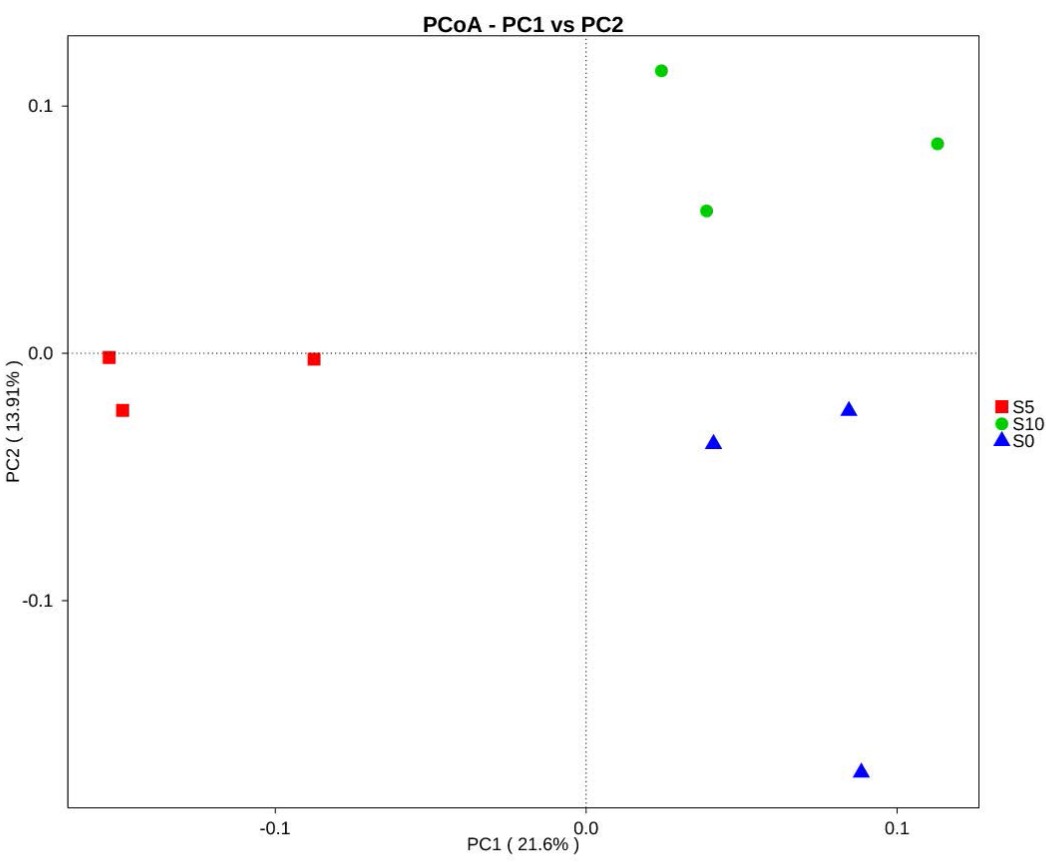

**Figure 5** **Principal coordinates analysis (PCoA) of the soil bacterial community composition based on Bray–Curtis distances.** S0, control; S5, normal treatment; S10 double treatment.

Deltaproteobacteria were primarily different in the high-litter treatment (double). The green color in Fig. 6 indicates the significantly different taxa in the normal treatment, and these species could potentially be used as biomarkers in the normal quantity treatment (Fig. 6).

Pearson correlation analysis showed that soil moisture, DON and MBN were the factors that mainly contributed to the significant correlation with bacterial taxa (Table 2). DON was significantly correlated with the relative abundance of Actinobacteria, Bacteroidetes, Verrucomicrobia, Verrucomicrobia, Firmicutes and Nitrospirae, with coefficients of $-0.684$, $0.812$, $0.679$, $0.669$, $-0.804$ and $0.715$, respectively. The SM and MBN were similarly correlated with the bacterial community composition (Table 2). There were no significant correlations with the relative abundance of Acidobacteria, Gemmatimonadetes and Chloroflexi, as the abundance of these taxa was stable among the different treatments.

## DISCUSSION

Plant litter decomposition is a key process of in the recycling of soil elements (*Berg & Mcclaugherty, 2014*). In this study, the SOC and STN contents were not significantly altered by litter decomposition (Fig. 2). This result is not consistent with other litter

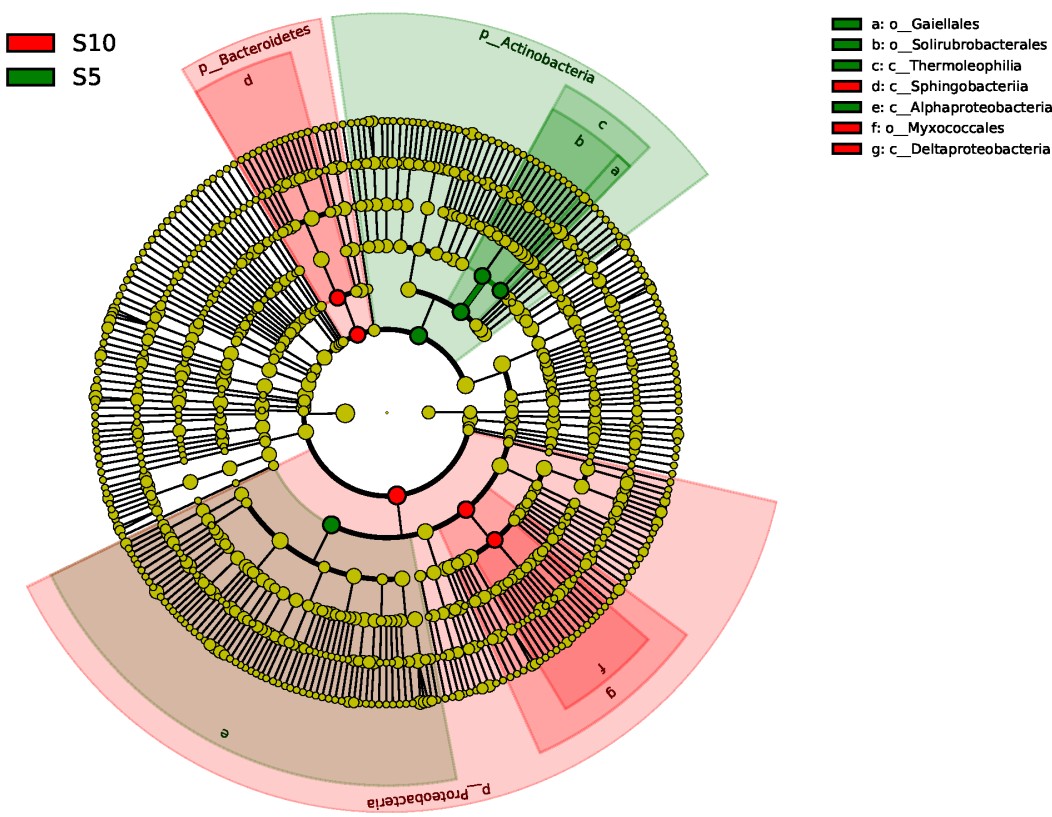

**Figure 6** A linear discriminant analysis effect size (LEsFe) method identifies the significantly different abundant taxa of bacteria under different litter quantity treatments. Taxa with significantly different abundance among treatments are represented by colored dots. S5, normal treatment; S10 double treatment.

**Table 2** The Pearson correlations between the soil properties and the soil bacterial community composition.

|  | DOC | DON | MBC | MBN | SM | NO$_3$-N | NH$_4$-N |
|---|---|---|---|---|---|---|---|
| Proteobacteria | **0.759**[*] | 0.302 | −0.227 | −0.426 | **0.676**[*] | 0.511 | −0.313 |
| Actinobacteria | −0.648 | **−0.684**[*] | −0.189 | **0.816**[**] | **−0.839**[**] | −0.444 | 0.514 |
| Bacteroidetes | 0.644 | **0.812**[*] | 0.33 | **−0.915**[**] | **0.749**[*] | 0.26 | −0.306 |
| Verrucomicrobia | 0.114 | **0.679**[*] | 0.511 | **−0.674**[*] | 0.385 | −0.035 | −0.343 |
| Verrucomicrobia | 0.537 | **0.669**[*] | 0.201 | **−0.785**[*] | **0.674**[*] | 0.395 | −0.462 |
| Firmicutes | −0.623 | **−0.804**[**] | −0.262 | **0.897**[**] | **−0.820**[**] | −0.404 | 0.426 |
| Nitrospirae | 0.563 | **0.715**[*] | 0.307 | **−0.797**[*] | 0.637 | 0.239 | −0.318 |

**Notes.**

DOC, dissolve organic carbon; DON, dissolve organic nitrogen; MBC, microbial biomass carbon; MBN, microbial biomass nitrogen; SM, soil moisture; NO$_3$-N, nitrate nitrogen; NH$_4$-N, ammonia nitrogen.

[*]Indicate significance at the 0.05 level.

[**]Indicate significant at the 0.007 level (adjusted by Bonferroni correction).

decomposition studies. This study was a short-term experiment (only three months), generally, while total C and N accumulation in soil occurs over long term processes with different mechanisms. However, the available nutrients in soil, such as nitrite nitrogen and dissolved nitrogen, were significantly altered by litter decomposition. Litter decomposition altered the available soil N fractions (i.e., MBN, DON and $NO_3$-N), and provided N resources for the growth of microbial organisms (*Cleveland & Townsend, 2006*; *Wardle et al., 2004*). The MBC and DOC also differed between the different treatments. These changes revealed that the available C and N concentrations in the soil were sensitive to litter decomposition, which could help to estimate and evaluate the effects of litter decomposition under global climate change, N deposition, extreme drought and other environmental problems.

Litter decomposition altered the bacterial community composition by a greater degree in the normal quantity treatment than in the double treatment, but the bacterial diversity did not differ significantly (Shannon and observed-species indices). Short-term litter decomposition increased the relative abundance of Actinobacteria, Firmicutes and Thermoleophilia, and decreased the relative abundance of Deltaproteobacteria, Gammaproteobacteria, Betaproteobacteria and Sphingobacteriia, which is most likely a result of the available C and N input via litter deposition caused by soil or litter microorganisms (*Cleveland & Townsend, 2006*; *Wardle et al., 2004*). Soil copiotrophic Bacteroidetes, α-, β-, and γ-Proteobacteria were relatively more abundant in the control and the double quantity litter treatment soils. The available nutrients released by the litter stimulated the microbial production of extracellular enzymes (*Koyama et al., 2013*), resulting in increased C and N availability, which also altered the bacterial community composition. *Zhang et al. (2016)* (*Zhang et al., 2016*) also observed that soil Proteobacteria increased with succession in Loess Plateau grasslands, as the soil nutrients were enhanced across the succession. In addition, our results indicated that soil water content significantly increased with the quantity of litter (Table 1). Increased water availability should alter soil microbial processes such as litter decomposition and nutrient mineralization (*DeAngelis et al., 2015*). These results suggest that nutrient and water availability in the soil may help explain why the increase in litter input altered the soil bacterial community composition in the normal and control treatments.

Bacteria play an important role in the litter decomposition process. Most Alphaproteobacteria, Acidobacteria and Actinobacteria can degrade recalcitrant C in plant litter (*Barret, Morrissey & O'Gara, 2011*). Acidobacteria can grow on complex polymers, including plant hemicellulose or cellulose and fungal chitin (*Eichorst, Kuske & Schmidt, 2011*). With litter addition, the soil bacterial community composition changed. These changes were indicated between the control and the normal treatments. The cluster tree analysis, PCoA and one-way ANOSIM all indicated that double and control treatments had similar bacterial communities (Figs. 3, 5 and Table 3). These results were consistent with the results of the LEefSe analysis and taxa abundance. Based on the results of LEefSe analysis indicated that Gaiellaes, Solirubrobacterales, Thermoleophilia and Alphaproteobacteria were significantly different in the normal treatment, and Shphingobacteria, Myxococcales and Deltaproteobacteria were significantly different in double treatment, which suggested

**Table 3**  ANOSIM and SIMPER analysis between the different litter treatments.

| Group A & B | SIMPER Average similarity % | ANOSIM R value |
|---|---|---|
| Normal vs Double | 76.28 | 1 |
| Normal vs Control | 76.66 | 0.889 |
| Double vs Control | 81.02 | 0.296 |

that litter addition had significant effects on certain bacterial species (*Fanin, Hättenschwiler & Fromin, 2014*; *Mau et al., 2015*). The abundance of soil microbes was based on the nutritional preferences and functions of the microbes (*Banerjee et al., 2016*; *Mau et al., 2015*). The normal amount of litter addition altered the priming effects of soil bacterial communities, which has been confirmed by other researchers (*Banerjee et al., 2016*). Litter addition enhanced the decomposition of soil organic matter and altered the abundance of functional groups, as seen by the decline of copiotrophic bacteria. The double litter addition treatment did not alter the soil bacterial composition, as much more liable nutrients from litter decomposition could maintain the growth of copiotrophic bacteria.

Soil available nutrients may be the primary difference caused by these shifts. *Zhong, Yan & Shangguan (2015)* found that N addition caused changes of the soil bacterial and fungal communities in a long term field experiment. The SOC was another main factor that affected the affecting soil bacterial community composition. *Liu et al. (2014)* found that Actinobacteria was significantly positively related to SOC, and Deltaproteobacteria was significantly negatively related to SOC. However, similar results were not observed in this study, which was in accordance with the results from *Zhong, Yan & Shangguan (2015)*. We also found that soil total N had no significant effect on soil community structure, but soil available N was significantly related to the soil bacterial community. Soil available N is the main resource for soil bacterial growth, which caused the variation in soil bacterial community structure. *Zhang et al. (2016)* reported that the soil nitrate nitrogen content was significantly related to the soil bacterial community along a natural succession. *Yao et al. (2014)* found that the soil ammonium nitrogen content played an important role in the soil bacterial community compositions in the grass land soils of China. *Yuan et al. (2014)* also observed similar results in soil on the Tibetan Plateau. All these results confirmed that soil available N content was the main factor that drove these changes in the soil bacterial communities.

## CONCLUSION

These results suggested that normal litter quantity could alter soil bacterial community compositions. A higher quantity of litter did not affect the soil microbial community. Beta, Gamma, and Deltaproteobacteria were significantly decreased in the normal quantity litter addition treatment, and subsequently increased in the double quantity litter addition treatment. The bacterial communities transitioned from Proteobacteria-dominant (Beta-, Gamma-, and Delta) to Actinobacteria-dominant during decomposition of the normal quantity of litter. The soil available nutrients and the soil copiotrophic bacterial

communities were higher in the control and the double quantity of litter decomposition treatments. These results suggested that litter addition affected the soil bacterial structure, and can provide guidance to manage vegetation restoration with the increase of litter quantity.

### Funding

This study was supported by the National Natural Science Foundation of China (41671280), the Non-profit Industry Research Project of Chinese Ministry of Water Resources (201501045), Key Projects in the National Science & Technology Pillar Program during the Twelfth Five-year Plan Period (2015BAC01B01) and Special-Funds of Scientific Research Programs of State Key Laboratory of Soil Erosion and Dryland Farming on the Loess Plateau (A314021403-C6). The funders had no role in study design, data collection and analysis, decision to publish, or preparation of the manuscript.

### Grant Disclosures

The following grant information was disclosed by the authors:
National Natural Science Foundation of China: 41671280.
Chinese Ministry of Water Resources: 201501045.
National Science & Technology Pillar Program during the Twelfth Five-year Plan Period: 2015BAC01B01.
Scientific Research Programs of State Key Laboratory of Soil Erosion and Dryland Farming on the Loess Plateau: A314021403-C6.

### Competing Interests

The authors declare there are no competing interests.

### Author Contributions

- Quanchao Zeng, Yang Liu and Shaoshan An conceived and designed the experiments, performed the experiments, analyzed the data, contributed reagents/materials/analysis tools, wrote the paper, prepared figures and/or tables, reviewed drafts of the paper.

### Data Availability

NCBI SRP107086
https://trace.ncbi.nlm.nih.gov/Traces/sra/?run=SRR5563030.

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
