# Peer review of "Impact of litter quantity on the soil bacteria community during the decomposition of Quercus wutaishanica litter"

_PeerJ, doi:10.7717/peerj.3777_

## Round 0.1 · original submission · Major Revisions

Both reviewers have recommended that revisions are required, and the first reviewer has had significant concerns on the English writing as well. I have read the ms and the reviewers' comments, and decided that major revisions must be made before it could be considered for publication.

Reviewer 1 ·

Basic reporting

It was shown in "General comments for the author"

Experimental design

It was shown in "General comments for the author"

Validity of the findings

It was shown in "General comments for the author"

Additional comments

The manuscript by Zeng et al., presents the results of the impact of litter quantity on soil bacteria community in the litter decomposition of Quercus wutaishanica. An incubation experiment under different litter quantity (normal quantity, 200 g/(m2.yr); double quantity, 400 g/(m2.yr) and control, none litter) was conducted. The results showed that litter quantity had significant effects on soil carbon fractions, nitrogen fractions, and bacterial community compositions, but no significant effects on soil bacterial diversity. Bacterial communities transitioned from Proteobacteria-dominant (Beta-, Gamma-, and Delta) to Actinobacteria-dominant during the litter decomposition with normal quantity. In general, the manuscript was well written and presented interesting results, but has several concerns about it that need to be addressed in the revision.
1. Introduction: the importance of this work was not present clearly, the author should reference more recent literature and point out the significance of the work.
2. Material and Methods: 1. why do you chose the 114 normal quantity (200 g/(m2•yr)) litters, double quantity (400 g/(m2•yr)) litters, what do you according to? 2. Line 123: pyrosequencing? in your sequencing, it was used by Illumina HiSeq 2500 platform.3. in the bacterial diverisity calculation, you used six indices: Observed-species, Chao1, Shannon, Simpson, ACE, Good-coverage. I suggest you can delete 2-3 indices because repeated information.
3. Results: the result is redundant now, and it should be refined.
4. Discussion: Discussion part should be greatly improved and major revision should made, including consult more relevant literature, comparison with results from other studies, and the meaning of your findings.

I recommend making a revision for this manuscript.

Reviewer 2 ·

Basic reporting

The manuscript „Impact of litter quantity on soil bacterial community in the litter decomposition of Quercus wutaishanica “ authored by Zeng and colleagues, deals with an interesting, but often time neglected aspect of litter decomposition in terrestrial forest ecosystems – the influence of litter quantity on the assemblage and structure of soil microbial communities. In a microcosm model experiment the authors used soil and Quercus wutaishanica litter collected at the Huangtu Plateau in China.


While the idea and general hypothesis are interesting, it is hardly possible to fully evaluate the presented manuscript, because the text is not written in a clear and professional English language. I urgently suggest that manuscript is checked by an English native speaking colleague or a professional language editing service.

Some examples:

L44: “affecting/affect” has to be changed to effecting/effect throughout the whole text

L62: “soil organisms was had been proved”

L 209: “… showed decreased at the normal…”

L240/241: Since simple past is used all the time for everything, it is quite hard to distinguish between knowledge from other papers and the conclusions the authors draw from their data. This is especially problematic in the whole discussion.

The scientific content of the manuscript is hardly evaluateable at the moment. Hence, I had the impression that the language barrier/ issues hampers a clear decision on the manuscript.

Experimental design

However, I provided some comments which should be addressed as well while revising the language.

L68: change “Grain for Grain” to “Grain for Green”
L71-73: The part about the litter decomposition and global C cycles are a little overstated – the manuscript is about a microcosm experiment (nothing more, but also nothing less!)
L85:L102: The detailed description of the site is too detailed, since no field experiments were performed and the soil and litter was only collected there. Hence it should be enough to write this part in a shorted/ more condensed way.
L119: Please define “humid environment”
L125-L136: The description of the “analysis of soil properties” is really extensive. Since only standard measurements were used, I would suggest to write it up shorter or to reference to the authors own, recent PLOS ONE paper.
L163: Why did the authors use so many different indices to access alpha-diversity? In fact some of the metrics used are no real estimators for diversity, e.g. Chao 1. I would suggest to use the observed number of species and the effective Shannon diversity exp(H’) see: Hill MO. Diversity and evenness: a unifying notion and its consequences. Ecology. 1973;54:427–432 or Lou Jost, 2006. Entropy and diversity. Oikos 113:363–375.
L165: The authors refer to beta diversity here, however the way they define it is not correct. A correct definition would be e.g. “Beta diversity can be defined as the variability in species composition among sampling units for a given area” (Anderson et al. 2006 Ecology Letters 9, 683-693). Furthermore, beta diversity is mentioned in the methods, but never accessed statistically through the manuscript. Hence this should be deleted or an additional analysis needs to be performed.
L177: What exactly is a “SKN multiple range comparison” -> Please comment.
L179: Pearson relation should be Pearson correlation.
L196: Please also report the non significant test results and also include the number of sequences per sample as an estimate for the abundance.
L226: You cannot conclude that double and control had the same bacterial community based on the analyses results presented before. I would suggest to run a PERMANOVA (Anderson 2001 A new method for non-parametric multivariate analysis of variance. Australian Ecology. 26 (1): 32–46.) on a Bray Curtis similarity matrix to statistically test the differences of the bacterial community composition among the treatments.
L228: “Biological relevance” -> I doubt that a LEfSe analysis can reveal “biological relevance” of species.
L323/333 and Table 2: Multiple correlations of bacterial groups and soil parameters are calculated. Since this is a multiple comparison procedure, the authors need to adjust for the accumulation of type I errors using e.g. the false discovery rate (Benjamini and Hochberg 1995. Controlling the false discovery rate: a practical and powerful approach to multiple testing. Journal of the Royal Statistical Society, Series B. 57 (1): 289–300) or more conservatively a Bonferroni correction.
Discussion and Figures 2/3/4: The authors do not really explain why the double and control treatments seem to be more similar than normal and double. Please comment or discuss in more detail.
Figure 2/3/4: Please state whether the error bars represent the standard deviation (SD) or the standard error (SE).
Table 1 is not properly formatted in the PDF version.

Validity of the findings

no comment

---

## Round 0.2 · Minor Revisions

The ms needs further minor revisions as recommended by one of the reviewers.

Reviewer 2 ·

Basic reporting

The authors of the manuscript “Impact of litter quantity on the soil bacteria community during the decomposition of Quercus wutaishanica litter” have done a great job and significantly improved the presentation of their study.

The article is now written a clear and professional English, sufficiently cites references and provides a solid background leading to the hypothesis of the study. The structure of the article is fine and all basic requirements are met. However, I would insist on revising Figure 4, because the insert is still hard to read and the text if basically not readable. There is one other little error in Tab 3. As you wrote in the manuscript the name of the analysis is SIMPER not PRIMER, please correct.

Hence, I would recommend publishing the manuscript in PeerJ after this minor error have been corrected.

Experimental design

no comment

Validity of the findings

no comment

---

## Round 0.3 · accepted · Accept

Thank you for addressing the final minor correction. The revised manuscript is now acceptable for publication in the journal.